# The autophagy protein ATG14 safeguards against unscheduled pyroptosis activation to enable embryo transport during early pregnancy

Pooja Popli[1], Arin K Oestreich[2,3], Vineet K Maurya[4], Marina N Rowen[2,3], Yong Zhang[5], Michael J Holtzman[5], Ramya Masand[1], John P Lydon[4], Shizuo Akira[6,7], Kelle Moley[2,3], Ramakrishna Kommagani[1,8]*

[1]Department of Pathology and Immunology, Baylor College of Medicine, Houston, United States; [2]Department Obstetrics and Gynecology, Washington University School of Medicine, St. Louis, United States; [3]Center for Reproductive Health Sciences, Washington University School of Medicine, St. Louis, United States; [4]Department of Molecular and Cellular Biology, Baylor College of Medicine, Houston, United States; [5]Department of Medicine and Department of Cell Biology, Washington University School of Medicine, St. Louis, United States; [6]Department of Host Defense, Research Institute for Microbial Diseases (RIMD), Osaka, Japan; [7]Laboratory of Host Defense, World Premier Institute Immunology Frontier Research Center (WPI-IFReC), Osaka University, Osaka, Japan; [8]Department of Molecular Virology and Microbiology, Baylor College of Medicine, Houston, United States

*For correspondence:
Rama.Kommagani@bcm.edu

Competing interest: The authors declare that no competing interests exist.

## eLife Assessment

This **important** study reports a novel function of ATG14 in preventing pyroptosis and inflammation in oviduct cells, thus allowing smooth transport of the early embryo to the uterus and implantation. The data supporting the main conclusion are **convincing**. This work will be of interest to reproductive biologists and physicians practicing reproductive medicine.

**Abstract** Recurrent pregnancy loss, characterized by two or more failed clinical pregnancies, poses a significant challenge to reproductive health. In addition to embryo quality and endometrial function, proper oviduct function is also essential for successful pregnancy establishment. Therefore, structural abnormalities or inflammation resulting from infection in the oviduct may impede the transport of embryos to the endometrium, thereby increasing the risk of miscarriage. However, our understanding of the biological processes that preserve the oviductal cellular structure and functional integrity is limited. Here, we report that autophagy-related protein ATG14 plays a crucial role in maintaining the cellular integrity of the oviduct by controlling inflammatory responses, thereby supporting efficient embryo transport. Specifically, the conditional depletion of the autophagy-related gene *Atg14* in the oviduct causes severe structural abnormalities compromising its cellular integrity, leading to the abnormal retention of embryos. Interestingly, the selective loss of *Atg14* in oviduct ciliary epithelial cells did not impact female fertility, highlighting the specificity of ATG14 function in distinct cell types within the oviduct. Mechanistically, loss of *Atg14* triggered unscheduled pyroptosis via altering the mitochondrial integrity, leading to inappropriate embryo retention and impeded embryo transport in the oviduct. Finally, pharmacological activation of pyroptosis in pregnant mice phenocopied the genetically induced defect and caused impairment in embryo

transport. Together, we found that ATG14 safeguards against unscheduled pyroptosis activation to enable embryo transport from the oviduct to uterus for the successful implantation. Of clinical significance, these findings provide possible insights into the underlying mechanism(s) of early pregnancy loss and might aid in developing novel prevention strategies using autophagy modulators.

## Introduction

A successful pregnancy is orchestrated by the sequential and coordinated events happening in the female reproductive tract (FRT) (*Wang and Dey, 2006*). Each of these events is crucial to advance to the next step in pregnancy. For example, as a first step, the ovary undergoes ovulation to release a mature ovum that is collected by oviduct fimbriae and funneled through the infundibulum into the ampulla where they get fertilized by sperm. The embryos then pass through the ampulla-isthmus junction to enter the isthmus of the oviduct before exiting via the distal utero-tubal junction (UTJ) to implant on the wall of the uterus. Recently, the intricate structural and cellular diversity of the oviduct has drawn significant attention, emphasizing its vital role in facilitating embryo development and transport (*Ford et al., 2021*; *McKey et al., 2022*; *Harwalkar et al., 2021*). Morphologically, the oviduct is divided into four evolutionarily conserved regions: the infundibulum (nearest the ovary), ampulla (the site of fertilization), isthmus (serving as a sperm reservoir and site for early embryonic development), and UTJ (connected to the uterus). Each of these segments comprises epithelial, stromal, and smooth muscle cells with lumen lined up with an epithelium containing secretory (PAX8$^+$) and multi-ciliated (FOXJ1$^+$) cells. While the infundibulum and ampulla are composed of more ciliated than secretory cells, the isthmus has more secretory than ciliated cells (*Dirksen and Satir, 1972*; *Li et al., 2017*). Successful pregnancy requires that the embryo transit the entire oviduct orchestrated by the following: (1) secretory cells producing oviduct fluid, (2) beating cilia to ensure unidirectional flow of the fluid, and (3) periodic contractions of surrounding muscle to propel fluid flow. Therefore, a combination of ciliary and muscular activity contributes to the overall success of oviduct embryo transport. This combination of ciliary and muscular activity plays a crucial role in the effective transport of the embryo through the oviduct. Any perturbations in these early pregnancy events can lead to adverse ripple effects that can compromise pregnancy outcomes. Previous studies have reported that impaired oviductal transport of embryos can lead to pregnancy failure and cause infertility in mice (*Jiang et al., 2022*; *Herrera et al., 2020*; *Wang et al., 2004*; *Qian et al., 2018*). However, the underlying mechanism is not yet completely understood.

Autophagy is a cellular process, evolutionarily conserved from yeast to mammals, that recycles long-lived proteins and organelles to maintain cell energy homeostasis. Autophagy is classically activated through the nutrient sensor or mammalian target of the rapamycin complex. Genetic screens for autophagy-defective mutants in yeast and other fungi have currently identified 41 autophagy-related (ATG) genes that play a primary role in autophagy (*Klionsky, 2007*; *Wen and Klionsky, 2016*). Approximately half of these genes have homologs in higher organisms, and *Atg14* (also known as Barkor for Beclin 1 [*Becn1*]-associated autophagy-related key regulator) is one of them (*Itakura et al., 2008*; *Sun et al., 2008*). ATG14 is part of a protein complex that is composed of Beclin 1, vacuolar sorting protein 15 (VPS15), and VPS34 (also named as Pik3c3, the catalytic subunit of the class III phosphatidylinositol 3-kinase), and this ATG14-containing complex plays an important role in the initiation process of autophagy.

Pyroptosis is a type of inflammatory cell death mediated by gasdermin (GSDM) and is a product of continuous cell expansion until the cytomembrane ruptures, resulting in the release of cellular contents that can activate strong inflammatory and immune responses. GSDM family proteins are the primary executioners of pyroptosis. Cytotoxic N-terminal of GSDMs generated from caspase-mediated cleavage of GSDM proteins oligomerizes and forms pores across the cell membrane, leading to the release of proinflammatory cytokines such as interleukin-1β (IL-1β) and interleukin-18 (IL-18). Aberrant activation of pyroptosis has been implicated in the progression of many diseases, including cancer and autoimmune, cardiovascular, and infectious diseases (*Wang et al., 2019a*; *Man et al., 2017*; *Wei et al., 2022*). Recent studies have shown that the interactions between pyroptosis and autophagy play an important role in various physiological and pathological processes (*Wang et al., 2019a*; *Man et al., 2017*; *Wei et al., 2022*; *Lin et al., 2021*; *Li et al., 2024*; *Zhao et al., 2022*). For example, autophagy inhibition upregulates galangin-induced pyroptosis in human glioblastoma

multiforme cells and promotes pneumococcus-induced pyroptosis (**Kong et al., 2019**; **Kim et al., 2015**). However, the studies defining the molecular interactions between the pyroptosis, and autophagy pathways are very limited.

With a better understanding of the fundamental process of autophagy, its pathophysiological functions have begun to be appreciated in the FRT. For example, mice deficient in key autophagy genes such as genetic knockout of *Atg7* or *Becn1* result in primary ovarian insufficiency and reduced progesterone production (**Song et al., 2015**; **Gawriluk et al., 2014**). Similarly, recent studies from our group established the roles of three different autophagy-specific genes: *Atg16l*, *Fip200/Rb1cc1*, or *Becn1* in endometrial physiological processes, including receptivity and decidualization (**Oestreich et al., 2020a**; **Popli et al., 2023**; **Oestreich et al., 2020b**). However, none of these genes exhibited any discernible impact on oviduct function. Surprisingly, in this study, we revealed a critical role for ATG14 in maintaining proper oviduct function, specifically enabling the transport of embryos to the uterus—a function distinct from that of other autophagy-related proteins. Loss of ATG14 in the oviduct resulted in severe structural abnormalities, compromising its cellular integrity, and ultimately leading to embryo retention and infertility.

## Results

### Conditional deletion of *Atg14* in the FRT results in infertility despite the normal ovarian function

To explore the role of *Atg14* in uterine function, we first determined its expression levels in the uterus during early pregnancy (days 1–7) in mice. We found distinct expression of ATG14 in all the uterine compartments (luminal epithelium, glands, and stroma) by day 1 of pregnancy, which disappeared by day 2 of pregnancy (**Figure 1A**). However, the ATG14 expression in the uterus reappeared by day 3 and persisted through day 7 of pregnancy. The period from day 3 to day 7 is critical as it is during this time when the uterus begins to prepare for embryo implantation and undergoes the decidualization process. Consistent with protein expression, similar *Atg14* expression at mRNA levels was noted in uteri from early pregnant mice (**Figure 1B**). This analysis suggests a potential role for ATG14 protein in the uterine physiologic adaptations during early pregnancy. Thus, to study the role of ATG14 in uterine function, we generated a conditional knockout (*Pr*cre/+; *Atg14*flox/flox) mouse model by crossing *Atg14*flox/flox mice with mice expressing Cre recombinase under the control of progesterone receptor promoter (*Pr*cre/+). Histological examination of the uterus from adult females showed no gross morphological differences between *Atg14* cKO and control mice (**Figure 1—figure supplement 1A**). Further, we did not find any overt defects in the ovary as cKO mice had normal follicles and corpus luteum like their corresponding controls (**Figure 1—figure supplement 1B**). Analysis of transcript levels from the uterus, ovary, and liver showed that while *Atg14* levels were efficiently depleted in uteri from cKO mice the levels were unaltered in the ovary and liver samples in control and cKO mice (**Figure 1C**). Immunofluorescence analysis further confirmed the efficient deletion of ATG14 in all uterine compartments (epithelium and stroma) of cKO mice uteri compared to corresponding control groups (**Figure 1D**).

Considering the effective ablation, a 6-month breeding study was performed mating virile wild-type male mice with adult *Atg14* cKO and control female mice. We found that *Atg14* cKO females did not deliver any litter during the 6-month trial period. However, the control female (*Atg14*flox/flox) mice delivered an average of ~7–8 pups per litter every month (**Table 1**; **Figure 1E**). These findings suggest an indispensable role of *Atg14* in female fertility with intact ovarian function.

### Loss of *Atg14* results in impaired embryo implantation and uterine receptivity in mice

Based on the normal ovarian morphology, we posited that the infertility observed in females with *Atg14* cKO status could be attributed to compromised uterine functioning. Analysis of the implanting embryos at day 5 of pregnancy showed no implantation sites in the uteri of *Atg14* cKO females, whereas ~8–9 implantation sites were seen in the uteri of control mice at 5 days post coitum (dpc) (**Figure 2A**, left panel). While there was no embryo present in the uterine lumen of *Atg14* cKO mice, a fully attached embryo encapsulated by the luminal uterine epithelium was seen in control mice uteri (**Figure 2A**, middle panel). Additionally, MUC1, a receptive marker expression, persisted in the luminal epithelium of cKO mice uteri at 5 dpc as well as 4 dpc (**Figure 2A**, right panel, **Figure 2C**). The

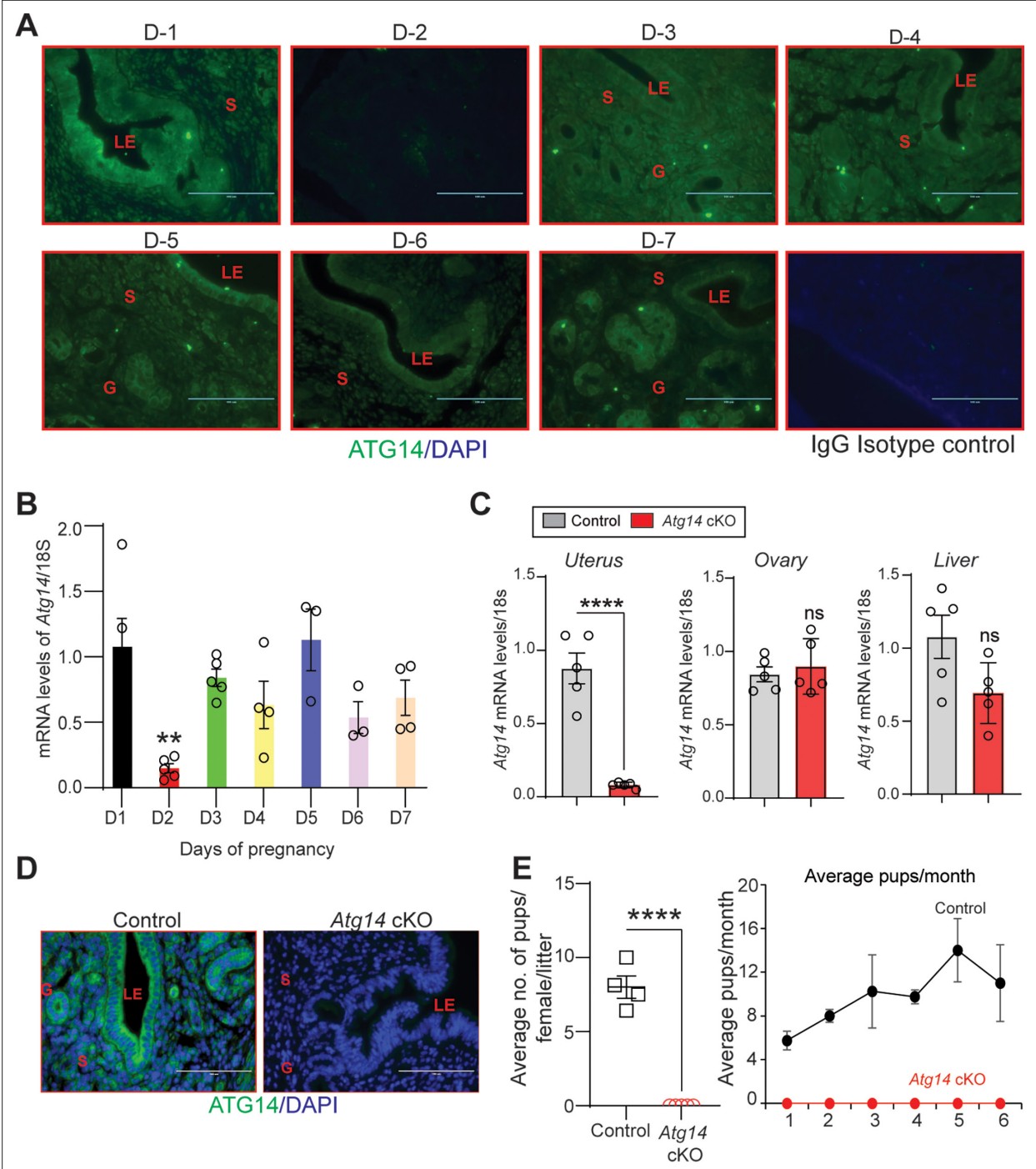

**Figure 1.** *Atg14* loss in the female reproductive tract (FRT) results in infertility. (**A**) Representative immunofluorescent images of uteri from pregnant mice (n = 3) at the indicated days of pregnancy stained with an ATG14-specific antibody (green). LE: luminal epithelium; G: glands; S: stroma, Scale bar: 100 μm. Rabbit IgG was used as an isotype control for staining. (**B**) Relative transcript levels of *Atg14* mRNA in uteri from pregnant mice (n = 3–5) at indicated days of pregnancy. mRNA levels are normalized to levels of 18S m-RNA. Data are presented as mean ± SEM; **p<0.01, p>0.05, ns = nonsignificant. (**C**) Relative mRNA levels of *Atg14* in 8-week-old virgin control and cKO mice uteri, ovary, and liver (n = 5). mRNA levels are normalized to levels of 18S mRNA. Data are presented as mean ± SEM; ***p<0.001, p>0.05, ns = nonsignificant. (**D**) Representative immunofluorescent images of ATG14 expression in different uterine compartments in control (n = 5) and *Atg14* cKO mice (n = 5). LE: luminal epithelium; G: glands; S: stroma. (**E**) (Left panel) Relative number of pups/female/litter and (*right panel*) and relative number of total pups/months of *Atg14* control (n = 4) and cKO mice (n = 5) sacrificed after the breeding trial. Data are presented as mean ± SEM; ***p<0.001; p>0.05, ns = nonsignificant.

The online version of this article includes the following figure supplement(s) for figure 1:

**Figure supplement 1.** *Atg14* cKO mice show normal gross morphology of the uterus and ovary (**A, B**).

**Table 1.** Six-month breeding trial of *Atg14* control and cKO females with wild-type males.

| Genotype | Females | Pups | Pups/female | Litters | Pups/litter |
|---|---|---|---|---|---|
| Control | n = 4 | 223 | 55.75 ± 7.32 | 28 | 7.96 ± 0.37 |
| *Atg14* cKO | n = 6 | 0 | 0 | 0 | 0 |

process of successful embryo attachment and implantation within the uterus necessitates a transition from a non-receptive to a receptive state, a transformation orchestrated under the regulated influence of steroid hormones. Thus, the effects of *Atg14* loss on uterine responsiveness to steroid hormones E2 and P4 were characterized. We performed an established hormone-induced uterine receptivity experiment, which involved ordered and co-stimulatory actions of E2 and P4, leading to the initiation of a receptive phase (**Tong and Pollard, 1999**). In response to E2 treatment, uteri from both *Atg14* control and cKO mice showed similar proliferation as evidenced by Ki-67 staining (**Figure 2B**, middle panel). Following the co-stimulation with E2 + P4 treatment, epithelial proliferation was inhibited

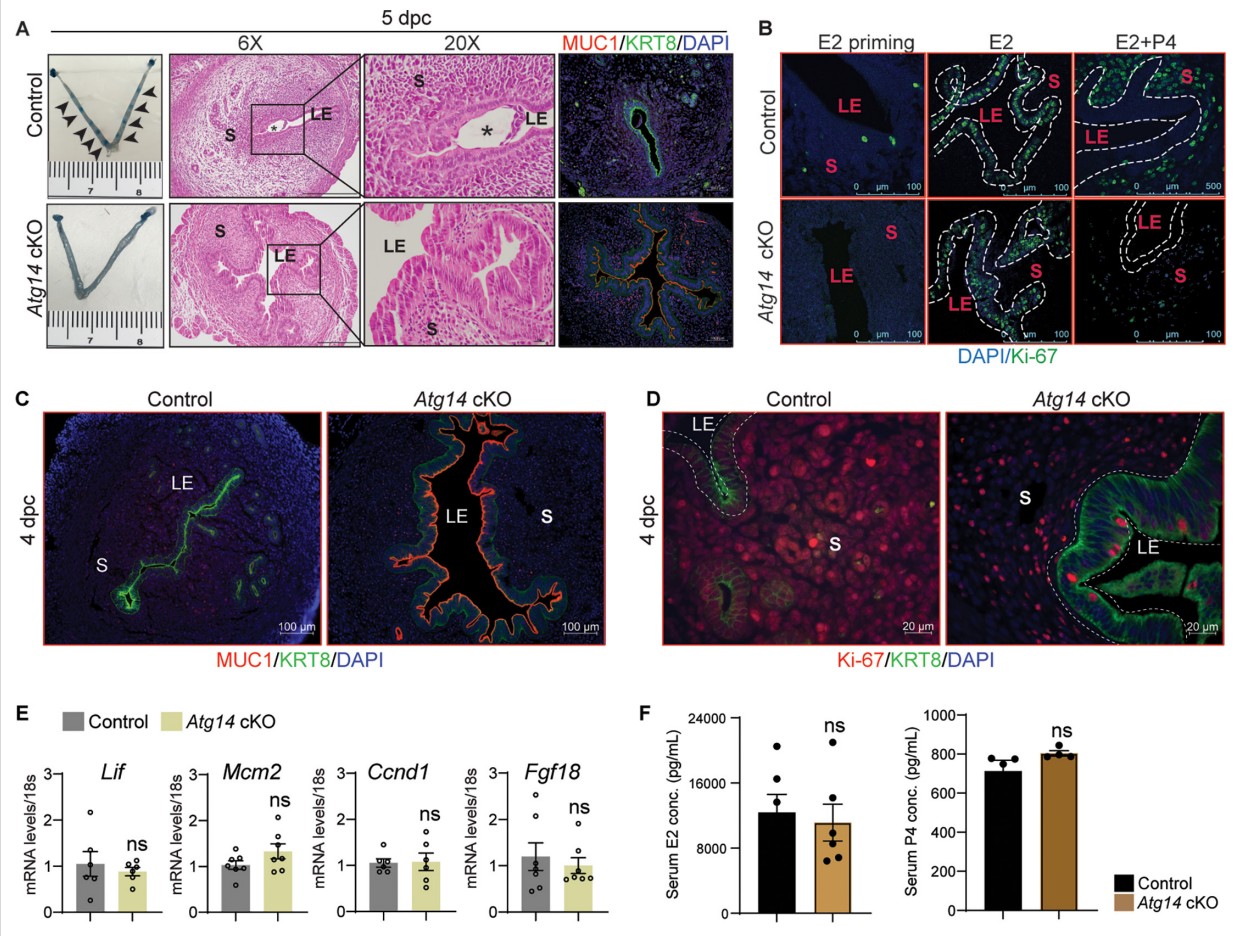

**Figure 2.** *Atg14* is critical for embryo implantation and uterine receptivity. (**A**) Gross images of 5.0 dpc uteri of control (n = 5) and *Atg14* cKO mice (n = 5) injected with Chicago Sky Blue dye to visualize implantation sites (denoted by black arrows) (left panel). H&E-stained cross-sections (×6 and ×20) of 5.0 dpc uteri of control (n = 5) and *Atg14* cKO (n = 5) mice to visualize embryo implantation (middle panel). The asterisk denotes the embryo. Immunofluorescence analysis of uterine tissues from control (n = 5) and *Atg14* cKO mice (n = 5), stained with MUC1 and KRT8 (right panel). LE: luminal epithelium; G: glands; S: stroma. (**B**) Representative immunofluorescence images of uteri from control (n = 5) and *Atg14* cKO mice (n = 5) stained for Ki-67 following Oil or E2 or E2 + P4 treatment (n = 5 mice/group); scale bar: 100 µm. LE: luminal epithelium; G: glands; S: stroma. (**C**) Immunofluorescence analysis of KRT8 (green), MUC1 (red), and (**D**) Ki-67 (red) in the uteri of 4 dpc control and *Atg14* cKO mice; scale bars, 100 µm. (**E**) Relative transcript levels of *Lif, Mcm2, Ccnd1,* and *Fgf18* in control and cKO uteri at 4 dpc. mRNA levels are normalized to levels of 18S m-RNA. Data are presented as mean ± SEM; ns = nonsignificant. (**F**) Levels of steroid hormones estradiol and progesterone from serum collected during euthanasia of 4 dpc control or cKO mice.

with concomitant induction of stromal cell proliferation in control mice uteri indicating fully receptive uteri (*Figure 2B*). Interestingly, cKO mice uteri failed to elicit sub-epithelial stromal cell proliferation and showed intact P4-driven inhibition of epithelial proliferation. Consistently, uteri from *Atg14* cKO mice at 4 dpc showed a reduced number of proliferating stromal cells (*Figure 2D*). The absence of significant changes in E2-induced targets, such as *Lif*, *Mcm2*, *Ccnd1*, and *Fgf18*, at 4 dpc (*Figure 2E*) supports our conclusion that ATG14 is required for P4-mediated but not for E2-mediated actions during uterine receptivity. Moreover, the normal serum levels of E2 and P4 at D4 of pregnancy rule out any hormonal imbalances, strongly suggesting that the observed phenotype is primarily due to the uterine-specific loss of *Atg14* (*Figure 2F*).

## *Atg14* is required for maintaining oviductal cell structural integrity and embryo transport

Given that *Atg14* cKO mice had impaired embryo implantation, we wondered whether embryos are reaching the uterus timely through the oviduct. To determine this, we flushed embryos from both the control and cKO mice uteri on day 4 of pregnancy. Interestingly, in cKO mice, we could retrieve only 1–2% of embryos from their uteri, whereas in control mice, 100% of well-developed blastocysts were retrieved from their uteri (*Figure 3A and B*). To ensure the timely transport of all embryos from the oviducts to the uteri, we also flushed oviducts from both control and cKO mice. As expected, in the control mice oviduct flushing, we could not recover any embryos or blastocysts, indicating their precise and timely transport to the uterus. Unexpectedly, oviduct flushing from cKO mice resulted in the retrieval of approximately 90% of embryos, suggesting their potential entrapment within the oviducts, impeding their transit to the uterus (*Figure 3A and B*). At 4 dpc, there was no significant difference in the average number of blastocysts, morula, or nonviable or retrieved from *Atg14* control uteri and cKO oviducts (*Figure 3—figure supplement 1*). However, we noted that the percentage of developmentally delayed embryos appeared to be higher in *Atg14* cKO oviducts compared to the embryos retrieved from control uteri (*Figure 3C*). The histological analysis further confirmed an entrapped embryo in the ampulla of *Atg14* cKO at 4 dpc as shown in *Figure 3D*. Given the embryo retention phenotype in oviduct, we sought to determine if ATG14 is expressed in this region. Consistent with previous studies reporting PR-cre activity in the isthmus (*Herrera et al., 2020*; *Soyal et al., 2005*), we found a significant depletion of ATG14 in the isthmus compared to the ampulla (*Figure 3—figure supplement 2A*). Further, the increased p62 and LC3B expression in cKO oviducts suggests that the observed embryo retention phenotype might be attributed to the loss of ATG14-dependent autophagy in cKO oviducts (*Figure 3—figure supplement 2B*).

Similarly, when we super-ovulated the mice and looked for the embryos on day 4 of pregnancy, we could recover ~20–25 embryos from the cKO mice oviducts compared to only 4–5 embryos that were able to reach the uterus. In comparison, in control mice, 90% of embryos were able to complete their predestined and timely transport to the uterus (*Figure 3E*), except 1–2% of unfertilized embryos, which remained in their oviducts.

To understand the underlying cause for retained embryos in cKO mice oviducts, we performed histology analysis and determined the structural morphology of their oviducts. The cKO mice oviduct lining shows marked eosinophilic cytoplasmic change akin to decidualization in human oviducts. Some of the cells showed degenerative changes with cytoplasmic vacuolization and nuclear pyknosis, loss of nuclear polarity, and loss of distinct cell borders giving an appearance of fusion of cells (*Figure 3F*). The marked cytologic enlargement appears to cause luminal obliteration or narrowing resulting in a completely unorganized, obstructed, and narrow lumen, thereby hampering the path of embryos to reach the uterus as evident from an entrapped embryo shown in *Figure 3D*. Further, epithelial (cytokeratin KRT8-positive) and myosalpinx (α-smooth muscle active [SMA]-positive) marker analysis revealed completely distorted epithelial structures (in terms of loss of epithelial cell integrity) with no overt defects in the muscle organization in oviducts of cKO mice (*Figure 3F*, lower panel). Notably, the ampullary region from cKO oviducts seems to be normal with intact epithelial and smooth muscle structures as evident from KRT8 and SMA staining (*Figure 3F*, upper panel).

## Selective loss of *Atg14* in oviduct cilia is dispensable for female fertility

The oviduct epithelium primarily consists of two types of cells: ciliated and secretory (non-ciliated) cells. Ciliated cells play a role in embryo and oocyte transport by means of ciliary beat and non-ciliated/

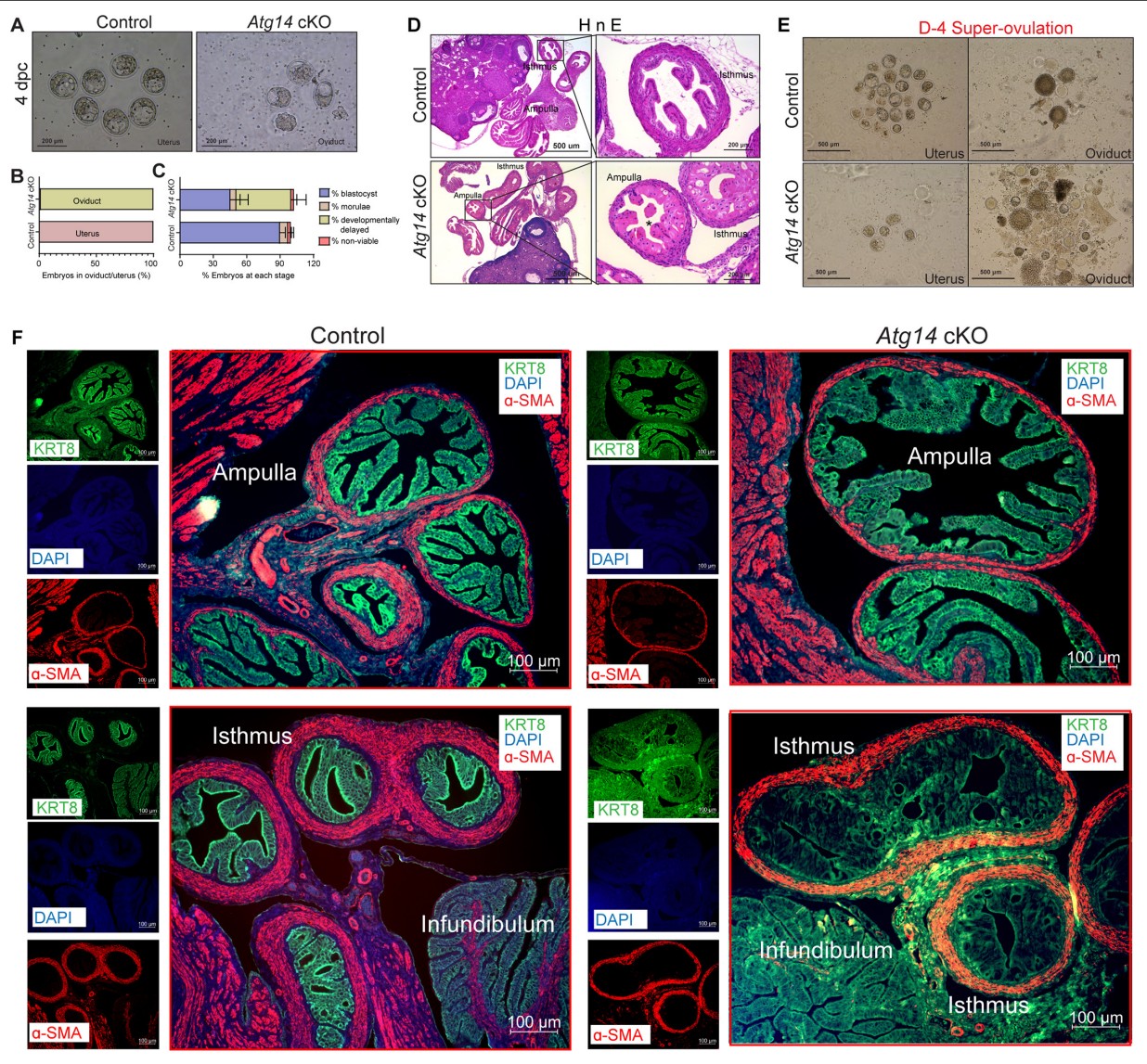

**Figure 3.** *Atg14* is critical for embryo transport in oviduct. (**A**) Representative images (upper panel) and (**B**) percentage of embryos (lower panel) collected at 4 dpc from the uteri or the oviducts of control (n = 6) or *Atg14* cKO mice (n = 6–8). (**C**) Percentage of blastocysts, morulae, developmentally delayed or nonviable embryos collected from *Atg14* control mice uteri and *Atg14* cKO mice oviducts at 4 dpc. (**D**) Histological analysis using H&E staining of the ampullary and isthmic region of the oviduct from control and *Atg14* cKO female mice at 4 dpc (n = 3 mice/genotype). (**E**) Embryos retrieved from the oviduct and uterus of super-ovulated *Atg14* control or cKO mice at 4 dpc (n = 3 mice/genotype). (**F**) Immunofluorescence analysis of KRT8 (green), and α-SMA (red), in the oviduct of 4 dpc control (n = 5) and *Atg14* cKO mice (n = 5) ampulla (upper panel) and isthmus (lower panel).

The online version of this article includes the following source data and figure supplement(s) for figure 3:

**Figure supplement 1.** Average number of embryos are unaltered in control or *Atg14* cKO D4 pregnant mice.

**Figure supplement 2.** *Atg14* cKO mice altered autophagy markers expression.

**Figure supplement 2—source data 1.** Original western blots for *Figure 3—figure supplement 2*, indicating the relevant bands and groups.

**Figure supplement 2—source data 2.** Uncropped original labelled blots.

secretory cells produce an oviductal fluid that is rich in amino acids and various molecules, thereby providing an optimal microenvironment for sperm capacitation, fertilization, embryonic survival, and development (*Ito et al., 2020*). Therefore, we determined whether oviducts from cKO mice possess normal ciliated and secretory cell composition. To do so, we examined the expression of various markers, including acetylated-α-tubulin (cilia marker), FOXJ1 (ciliogenesis markers), and PAX8 (non-ciliated/secretory cell marker), in control and cKO oviducts. As shown in *Figure 4A*, normal ciliary

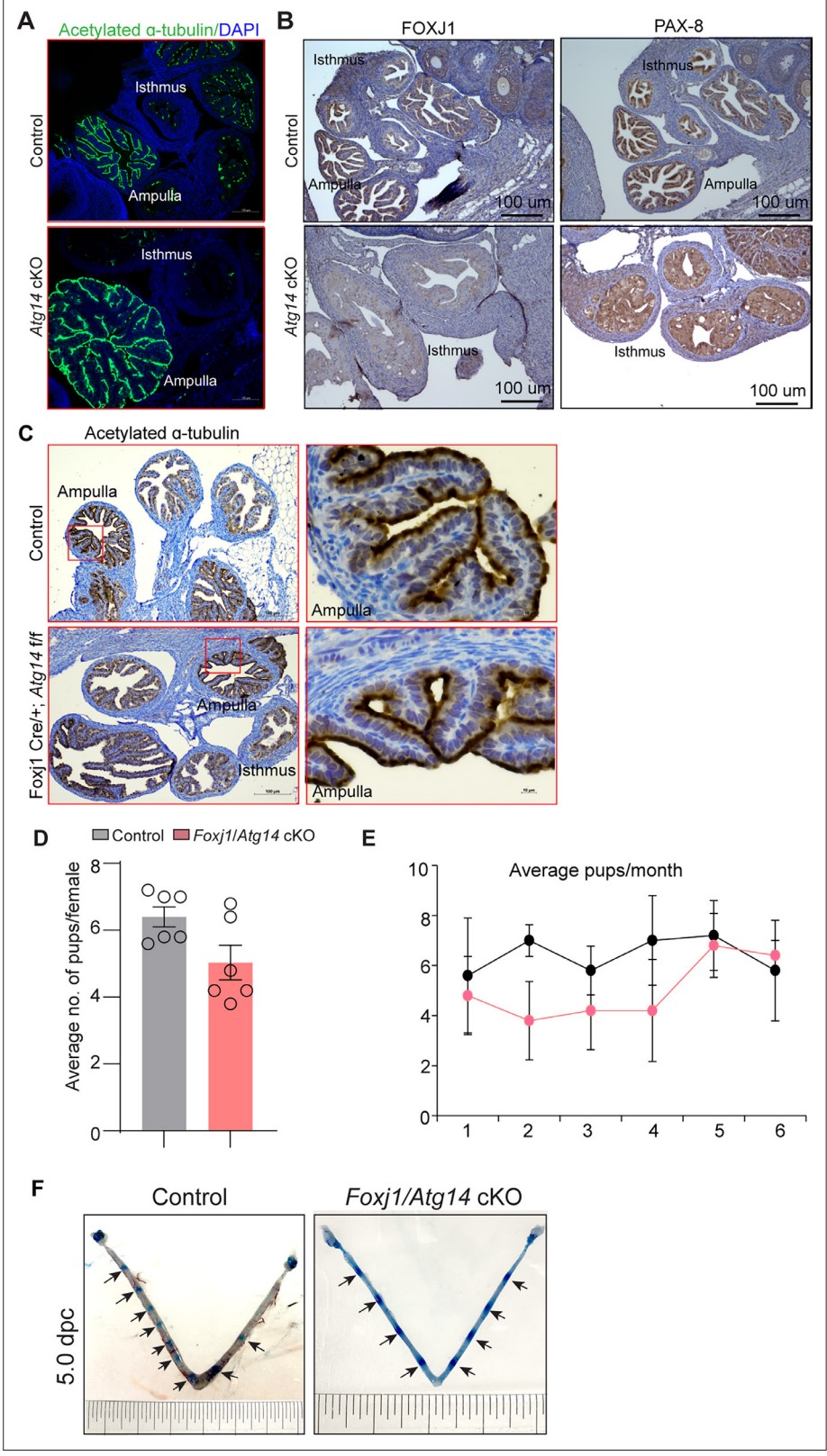

**Figure 4.** *Atg14* loss in oviduct cilia is dispensable for embryo transport. (**A**) Immunofluorescence analysis of acetylated α-tubulin (green), and DAPI (blue) in oviduct of 4 dpc control (n = 5) and *Atg14* cKO mice (n = 5). (**B**) Immunohistochemical analysis of FOXJ1 and PAX8 at 4 dpc (n = 5). (**C**) Immunohistochemical analysis of acetylated α-tubulin in 8-week-old control (n = 5) and *Foxj1/Atg14* cKO mice (n = 5). Images are taken at ×6 and ×40. Scale

*Figure 4 continued on next page*

*Figure 4 continued*

bar: 100 µm and 10 µm. (**D, E**) Relative number of pups/females/litter and relative number of total pups/months of control (n = 5) and *Foxj1/Atg14* cKO mice (n = 5) sacrificed after the breeding trial. Data are presented as mean ± SEM; p>0.05, ns = nonsignificant. (**F**) Gross images of 5.0 dpc uteri of control (n = 3) and *Foxj1/Atg14 cKO mice* (n = 3) injected with Chicago Sky Blue dye to visualize implantation sites (denoted by black arrows).

The online version of this article includes the following figure supplement(s) for figure 4:

**Figure supplement 1.** *Atg14* cKO mice show a reduced number of FOXJ1-expressing ciliary epithelial cells.

structures were observed in the ampulla of both control and cKO oviducts. However, in the isthmus of cKO oviducts, we observed a reduction in both FOXJ1- and PAX8-expressing cells (**Figure 4B**, **Figure 4—figure supplement 1**). Given the importance of cilia in embryo transport, we wondered whether the loss of *Atg14* in oviduct cilia has any impact on embryo transport. To address this, we generated a *Foxj1*<sup>cre/+</sup>; *Atg14*<sup>flox/flox</sup> mouse model, wherein *Atg14* will be ablated only in ciliary epithelial cells. We observed that ciliated epithelial cells that were positive for acetylated α-tubulin staining did not appear to be different in *Foxj1*<sup>Cre/+</sup>; *Atg14*<sup>flox/flox</sup> mice oviduct compared to controls, suggesting normal ciliogenesis in *Foxj1*<sup>Cre/+</sup>; *Atg14*<sup>flox/flox</sup> mice (**Figure 4C**).

The 6-month breeding trial revealed that loss of *Atg14* in oviductal cilia had no impact on fertility (**Table 2**, **Figure 4D and E**). Consistently, D5 implantation study analysis showed ~7–8 visible embryo implantation sites in *Foxj1/Atg14* cKO mice like their corresponding controls, suggesting that embryos were able to make their way to the uterus in a timely manner and undergo implantation despite the ablation of *Atg14* in the oviduct ciliary epithelial cells (**Figure 4F**). These findings suggest that ciliary expression of *Atg14* is dispensable for embryo transport.

## ATG14 maintains mitochondria integrity and prevents unscheduled pyroptosis activation in the oviduct to enable embryo transport

To gain more insights into structural defects, we performed transmission electron microscopy (TEM) analysis on oviducts collected from both control and cKO females on day 4 of pregnancy. Interestingly, oviducts from cKO females had numerous altered mitochondrial structures with abnormally enlarged mitochondria and a less dense matrix compared to control oviducts that had small and compact mitochondria with tight cristae and a dense matrix (**Figure 5A**). Corroborating and extending those findings, we observed marked reductions of mitochondria network (as defined by TOM20 staining) in the sub-nuclear region of oviducts from *Atg14* cKO mice compared to more densely packed mitochondrial network in the sub-nuclear region of control ones (**Figure 5B**). Further co-localization study analysis revealed a uniform co-localization of cytochrome C with the TOM20-positive mitochondrial network in control oviducts reflecting the intact mitochondrial integrity (**Figure 5B**). In contrast, in *Atg14* cKO oviducts, cells with disrupted mitochondrial networks exhibited increased cytosolic leakage of cytochrome C (**Figure 5B**). Additionally, analysis of various mitochondrial functional and architecture markers (*Cox4i2*, *Pink1*, *Opa1,* and *Drp1)* showed reduced expression in *Atg14* cKO compared to controls (**Figure 5C**).

Having identified the cellular mechanisms of Atg14, we aimed to delineate the underlying molecular mechanisms associated with Atg14 function in the oviduct. Since we found mitochondrial defects with ATG14 loss and altered mitochondrial structures have been linked to the activation of pyroptosis, we determined if ATG14 regulates pyroptosis in the oviduct (**Liu et al., 2018**; **Shi et al., 2022**; **Wang et al., 2019b**). First, we assessed the key primary executors of the pyroptosis pathway, gasdermin D (GSDMD), and caspase-1. Immunofluorescence analysis revealed a remarkable upregulation in GSDMD and caspase-1 expression in the cKO oviducts compared to controls (**Figure 5D and E**). However, histological analysis of uterus and ovary showed no induction in GSDMD expression compared to their corresponding controls (**Figure 5F**). Western blot analysis further confirmed elevated expression

**Table 2.** Six-month breeding trial of *Foxj1/Atg14* control and cKO females with wild-type males.

| Genotype | Females | Pups | Pups/female | Litters | Pups/litter |
|---|---|---|---|---|---|
| Control | n = 5 | 192 | 38.4 ± 1.29 | 26 | 9.1 ± 1.29 |
| *Foxj1/Atg14* cKO | n = 5 | 151 | 30.2 ± 1.46 | 24 | 7.6 ± 1.46 |

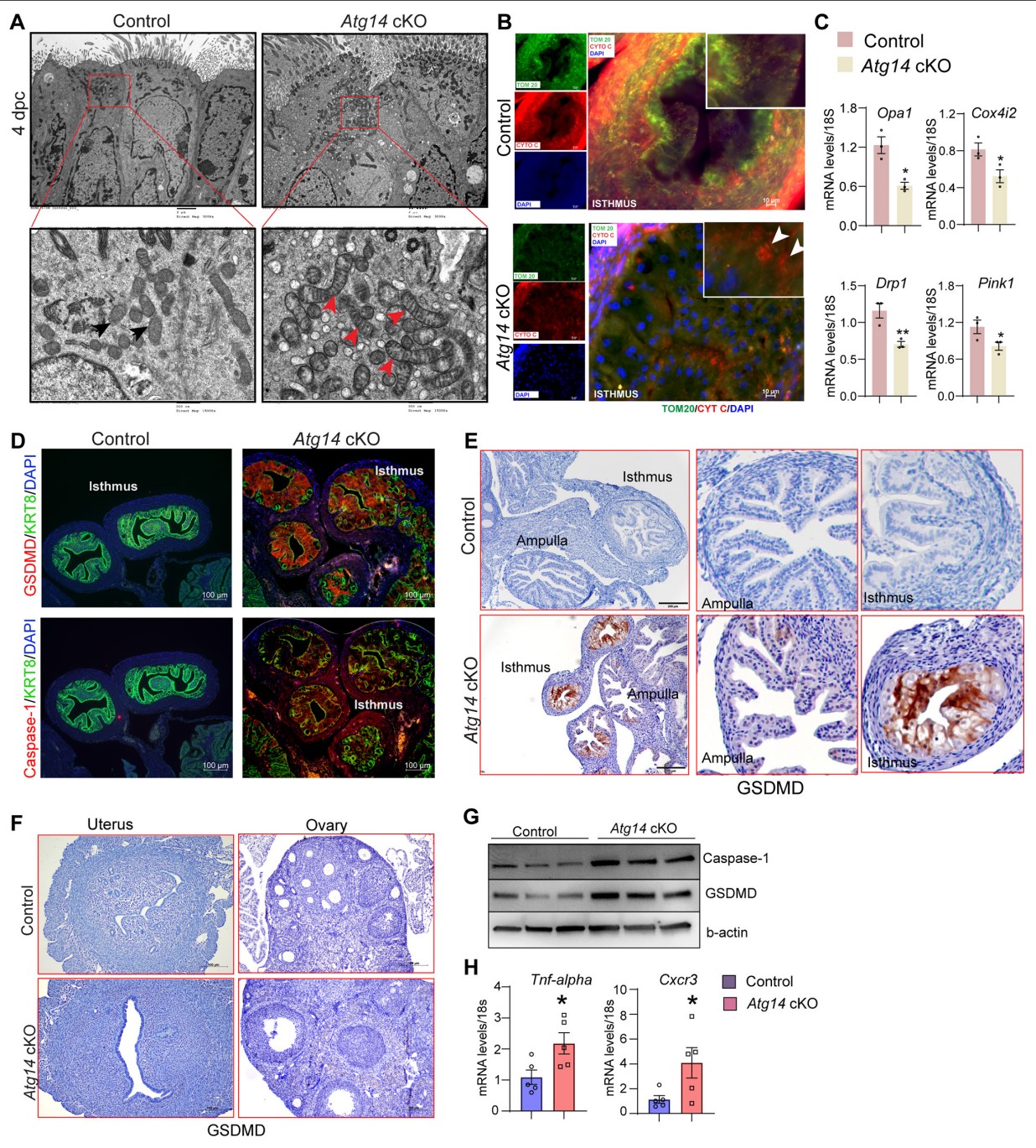

**Figure 5.** *Atg14* facilitates embryo transport in the oviduct by preserving mitochondrial integrity and inhibiting the activation of pyroptosis. (**A**) Transmission electron microscopy of oviducts at 4 dpc from control (n = 3) and *Atg14* cKO (n = 3). Black arrowheads in control oviducts show small, compact mitochondria with tight cristae. Red arrowheads in cKO oviducts show abnormally enlarged mitochondria with loose cristae. (**B**) Immunofluorescence analysis of TOM20 (green) and cytochrome C (red) in *Atg14* control and cKO oviducts cryo-sections. The inset shows the zoomed image of the selected area from control and cKO oviducts, Arrowheads in the cKO inset oviduct section show the leaked cytochrome C in the cytoplasm. Tissues were counterstained with DAPI (blue) to visualize nuclei; scale bars, 10 µm. (**C**) Relative transcript levels of *Opa1*, *Cox4i2*, *Drp1*, and Pink1 in oviduct tissues. Data are presented as mean ± SEM. *p<0.05; **p<0.01 compared with controls. 18S was used as an internal control. (**D**) Immunofluorescence analysis of GSDMD (red) + KRT8 (green), Caspase-1 (red) + KRT8 (green) in oviducts of adult control (n = 5), and *Atg14* cKO mice (n = 5). Tissues were counterstained with DAPI (blue) to visualize nuclei; scale bars, 100 µm. (**E**) Immunohistochemical analysis of GSDMD expression in adult oviduct tissues (left panel). The middle and right panels show the zoom-in images to show the relative GSDMD expression in the ampulla and isthmus section from control and cKO oviducts. (**F**) Immunohistochemical analysis of GSDMD expression in adult uterus and ovary tissues. (**G**) Western blotting to show protein levels of Caspase-1 and GSDMD in oviduct tissues. β-actin is used as a loading control. (**H**) Relative transcript levels of *Tnfa*

*Figure 5 continued on next page*

*Figure 5 continued*

and *Cxcr3* in oviduct tissues. Data are presented as mean ± SEM. *p<0.05; **p<0.01; ***p<0.001 compared with controls. 18S was used as an internal control.

The online version of this article includes the following source data for figure 5:

**Source data 1.** Original western blots for *Figure 6G*, indicating the relevant bands and groups.

**Source data 2.** Uncropped labelled raw western blots.

of caspase-1 and GSDMD in cKO oviducts in comparison to control oviducts as shown in *Figure 5G*. Additionally, the qPCR analysis demonstrated elevated levels of inflammatory markers, such as *Tnfa* and *Cxcr3*, in cKO oviducts compared to control ones (*Figure 5H*). Based on these findings, we posit that Atg14 plays a crucial role in regulating the pyroptotic pathway by preserving the mitochondrial structural and functional integrity in the oviduct.

To substantiate the notion, we also evaluated the impact of unscheduled activation of pyroptosis on embryo transport. To test this, we employed a pyroptosis inducer, polyphyllin VI, and chose the optimal dose based on the established studies (*Teng et al., 2020*). Wild-type females were mated with virile males and following the plug detection treated with polyphyllin VI for three consecutive days from 1 to 4 dpc (*Figure 6A*). We chose to treat mice from 1 to 4 dpc to activate unwarranted pyroptosis in oviduct during the embryo transport. Following the treatments, the oviducts, and uteri from both vehicle-treated and polyphyllin VI-treated were flushed. We found that pregnant females treated with polyphyllin VI showed ~50% embryo retention in the oviduct, whereas in the vehicle-treated group, no embryos were retained in the oviduct (*Figure 6A–C*). Further analysis of embryos retrieved from polyphyllin-treated oviducts showed more percentage of developmentally delayed and non-viable embryos compared to embryos recovered from control uteri (*Figure 6D*). The average number of embryos recovered from polyphyllin- and vehicle-treated mice was not significantly different (*Figure 6—figure supplement 1*). Histological analysis showed a marked induction in GSDMD expression compared to vehicle-treated mice oviducts (*Figure 6E*). However, precisely activating the unscheduled pyroptosis during the critical period of embryo transport is technically challenging. Nonetheless, our findings provide evidence that unscheduled pyroptosis adversely affects embryo transport through the oviduct. Taken together, these results demonstrate that *Atg14* safeguards pyroptosis activation in the oviduct and allows the smooth transport of embryos to the uterus (*Figure 6F*).

## Discussion

In this study, we delineated the essential role of ATG14 in maintaining the structural integrity of the oviduct by preventing pyroptosis, which enables smooth embryo transit during early pregnancy. Specifically, we dissected the tissue-specific functions of ATG14 in the FRT using Cre driver mice targeting the FRT. Given that PR-Cre expresses postnatally in different tissues of FRT such as corpus luteum, oviducts, and different cellular compartments (epithelial, stromal, and myometrium) of uteri (*Soyal et al., 2005*), we found that conditional ablation of ATG14-mediated functions in FRT resulted in infertility owing to hampered transport of embryos from the oviduct. Additionally, *Atg14* loss in uteri causes impaired embryo implantation and receptivity. Its ablation in the oviduct might lead to the activation of pyroptosis, an inflammatory-associated apoptosis pathway that causes the retention of embryos in the oviduct and prevents their timely transport to the uterus. However, the ovarian-specific functions were found to be unaffected despite the loss of *Atg14* in the corpus luteum.

The human endometrium is a complex dynamic tissue that undergoes sequential phases of proliferation and differentiation to support embryo implantation during the conceptive. However, in the absence of an implanting embryo, the endometrium sheds (menstruation) and initiates the regeneration process (*Popli et al., 2022*). Previous reports indicated that autophagy is modulated in the human endometrium during the menstrual cycle. For example, autophagy is altered during the proliferative and secretory phases of the menstrual cycle with its highest activity occurring during the secretory phase when the stroma is decidualized (*Choi et al., 2014*). Similarly, the level of autophagy was higher in postmenopausal human uterine epithelial cells compared to premenopausal uterine epithelial cells, indicating the onset of autophagy upon estrogen deprivation (*Zhou et al., 2020*). Although

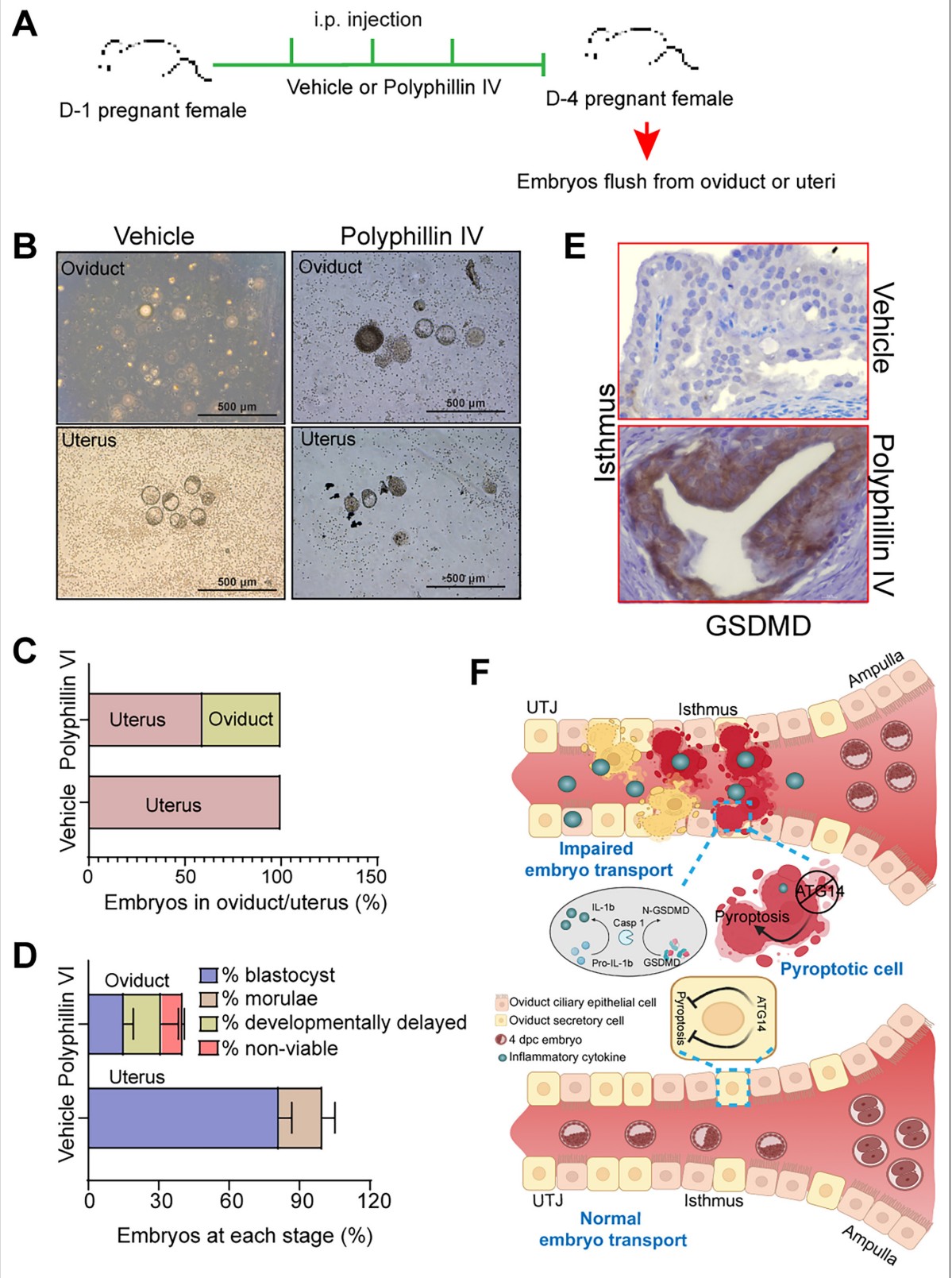

**Figure 6.** Pharmacological activation of pyroptosis in the oviduct inhibits embryo transport. (**A**) Experimental strategy for pyroptosis activation in pregnant female mice. (**B**) Embryos flushed from the vehicle (n = 3) or polyphyllin VI-treated (n = 3) D4 pregnant females. (**C**) Percentage of embryos recovered from oviducts or uteri. (**D**) Percentage of blastocysts, morulae, developmentally delayed or nonviable embryos collected from vehicle uteri

*Figure 6 continued on next page*

*Figure 6 continued*

or polyphyllin IV oviducts at 4 dpc. (**E**) Immunohistochemistry to show GSDMD expression in the isthmus section of polyphyllin IV-treated and vehicle-treated mice. ×40 objective, Scale bar: 5 µm. (**F**) Graphical illustration to show embryo transport and pyroptosis regulation in the oviduct.

The online version of this article includes the following figure supplement(s) for figure 6:

**Figure supplement 1.** Average number of embryos are unaltered in vehicle or polyphyllin VI-treated D4 pregnant mice.

all these studies reported that autophagy is hormonally regulated, however, the role of autophagy-specific proteins in the endometrium was not established till our group's recent reports. Specifically, studies from our group reported the role of three different autophagy-associated proteins: FIP200 (*Oestreich et al., 2020a*), ATG16L (*Oestreich et al., 2020b*), and BECLIN-1 in uterine functions (*Popli et al., 2023*). Conditional ablation of *Fip200* and *Atg16l* in the uterus displayed fertility defects owing to impaired implantation, uterine receptivity, and decidualization defects. On the contrary, loss of *Beclin1* in the uterus caused progressive loss of endometrial progenitor stem cells, resulting in severe uterine developmental defects and rendering the mice infertile (*Popli et al., 2023*). Although the autophagy-related proteins we studied so far influenced uterine functions (*Oestreich et al., 2020a*; *Popli et al., 2023*; *Oestreich et al., 2020b*), we found a distinct role for ATG14 in both the uterine and oviduct-specific functions. It is intriguing to note that the absence of ATG14 did not affect the tissue integrity of the uterus. However, the severe structural abnormalities in the oviduct due to *Atg14* ablation unearthed a unique function of Atg14 in maintaining oviduct homeostasis. This distinctive role of ATG14, unlike other autophagy proteins, might be due to its pivotal role in the assembly of PtdIns3K complexes, which is not the case for either FIP200 or ATG16L (*Popli et al., 2022*; *Fan et al., 2011*; *Diao et al., 2015*). Nevertheless, understanding the specific contributions of each core autophagy protein in reproductive tract functions is necessary that requires substantial efforts.

The retention of preimplantation embryos in the oviduct has been established as a significant contributor to implantation failures, presenting challenges to the overall reproductive health of women (*Herrera et al., 2020*; *Wang et al., 2004*; *Qian et al., 2018*). Embryo transport in the oviduct is known to be controlled primarily by two major physiological responses: ciliary activity and muscle contractility. In our study, a dramatic decrease in the number of Foxj1+ve ciliary epithelial cells and Pax-8+ve secretory cells in *Atg14* cKO mice implied cell-type-specific actions for ATG14 in the oviduct. Interestingly, the specific ablation of *Atg14* in Foxj1+ve ciliary epithelial cells of the oviduct does not appear to impact fertility. This suggests that the absence of ATG14 within ciliary cells does not impact the process of embryo transport. Although unexpected, this is consistent with other reports that have demonstrated the dispensability of cilia for embryo transport in the oviduct (*Herrera et al., 2020*; *Wang et al., 2004*).

Pyroptosis is a highly inflammatory form of programmed cell death, characterized by cell swelling, flattening of the cytoplasm, and large bubbles-like protrusions on the plasma membrane (*Ma et al., 2021*; *Cookson and Brennan, 2001*; *Jorgensen and Miao, 2015*). Several recent studies found a link between autophagy and pyroptosis (*Liu et al., 2021*). In vivo studies also found that mice lacking the autophagy genes *Atg14*, *Fip200*, *Atg5*, or *Atg7* in myeloid cells had more pronounced lung inflammation (*Lu et al., 2016*). In our study, cellular swelling and fused membranous structures (a unique feature of activation of pyroptosis) were observed in the oviductal epithelial folds from *Atg14* cKO mice. This aberrant activation of the pyroptosis pathway due to loss of *Atg14* appears to adversely affect the oviduct cellular structural integrity. These findings have implications beyond the oviduct as autophagy can modulate the inflammatory signaling pathway through different avenues. For example, first, autophagy prevents mitochondrial reactive oxygen species release that can activate the inflammasome by inhibiting IL1-β and IL-18 production through the digestion of dysfunctional mitochondria (*Netea-Maier et al., 2016*). Second, autophagy is capable of targeting inflammasome complexes for degradation, which prevents the cleavage of pro-IL-1β and pro-IL-18 into biologically active forms. Finally, autophagy machinery further regulates IL-1β levels by engulfing and degrading pro-IL-1β proteins (*Netea-Maier et al., 2016*). In our study, accumulation of abnormally enlarged and dysfunctional mitochondria in the absence of ATG14 in the oviduct clearly suggests the induction of severe inflammatory conditions hampering the embryo transit through the oviduct. These findings are further substantiated by the upregulation of pivotal mediators of the pyroptosis program, including GSDMD and caspase-1, in *Atg14* cKO mice. Notably, the induction of pyroptosis in the absence of ATG14 is specifically localized to the oviducts, with no parallel activation detected in either the

ovary or uterus. One potential reason for this observed phenomenon could be the oviduct's unique molecular signatures that get activated in response to the presence of gametes. A recent study by *Finnerty et al., 2024* reported that sperm induce pro-inflammatory conditions in the oviduct, which are preceded by an anti-inflammatory response triggered by the presence of embryos (*Finnerty et al., 2024*). The persistent activation of inflammatory or pyroptotic conditions following the loss of Atg14 suggests that Atg14 may be a critical autophagy protein responsible for suppressing pyroptosis in the oviduct. Moreover, understanding the complex rheostat between the ATG14 and inflammation regulatory axis is highly intriguing and will necessitate future studies employing sophisticated models, such as combined knockout mice where ATG14 is deleted alongside key inflammatory regulators (e.g., NLRP3, GSDMD, or CASPASE-1).

In this report, we revealed tissue-specific roles of ATG14 in governing oviductal transport, and uterine receptivity that are necessary for embryo implantation and pregnancy establishment. Of mechanistic insights, we delineated a novel function of ATG14 as a negative regulator of pyroptosis, which is vital for proper embryo transport in the oviduct (*Figure 6F*). Thus, this study not only sheds light on the involvement of autophagy in oviductal embryo transport but also underscores the importance of autophagy in preserving the oviduct tissue integrity. Such insights hold promise for advancing our understanding of gynecological pathologies associated with the oviduct including tubal pregnancies.

## Materials and methods
### Animal care and use
All animal studies were approved by the Institutional Animal Care and Use Committee of Washington University School of Medicine, Saint Louis, MO (protocol number: 20160227) and the Use Committee of Baylor College of Medicine, Houston, TX (protocol number: AN-8890). *Atg14*flox/flox mice were provided by a gift from Dr. Shizuo Akira at the Department of Host Defense, Research Institute for Microbial Diseases (RIMD), Osaka University, and previously described (*Matsunaga et al., 2009*). Wild-type Pr-cre mice were provided by Dr. John Lydon at Baylor College of Medicine, Houston, and previously described (*Soyal et al., 2005*). *Atg14*flox/flox mice, in which exon 4 was flanked by loxp sites, were bred to progesterone receptor cre (*Pr*cre/+) mice to generate (*Atg14*flox/flox; *Pr*cre/+ mice), hereafter referred to as *Atg14* cKO mice. Both control and conditional knockout females were generated by crossing females carrying homozygous *Atg14*flox/flox alleles with *Atg14* cKO males. *Foxj1-cre* mice were a generous gift from Dr. Michael J. Holtzman at Washington University St. Louis and previously described (*Zhang et al., 2007*). *Foxj1/Atg14* cKO mice were generated by crossing females carrying homozygous *Atg14*flox/flox females with *Atg14*flox/flox; *Foxj1*cre/+ males. All mice were age-matched and on a C57BL/6 genetic background (The Jackson Laboratory, Bar Harbor, ME). Mice were genotyped by PCR analysis of genomic DNA isolated from tail clippings using the gene-specific primers listed in *Supplementary file 1*.

### Fertility analysis and timed mating
Female fertility was determined by mating cohorts of *Atg14* cKO experimental (n = 6) and control *Atg14*flox/flox (n = 4) females individually starting at 8 weeks of age with sexually mature males of proven fertility. Similarly, breeding trials were set up for *Foxj1/Atg14*-cre females (n = 6) and *Foxj1/Atg14* control 8-week-old f/f females (n = 6). The numbers of litters and pups were tracked over 6 months for each female. Pups per litter for each genotype are reported as mean ± SEM. For timed mating, the morning on which the copulatory plug was first observed was considered 1 dpc. To visualize implantation sites, mice received a tail vein injection of 50 µL of 1% Chicago Sky Blue dye (Sigma-Aldrich, St. Louis, MO) at 5 dpc just before sacrifice.

### Steroid hormone treatments
The hormonal profile of pregnancy at the time of implantation was done using a previously described experimental scheme (*Popli et al., 2023*). Briefly, *Atg14* cKO and control females (8 weeks old) were bilaterally ovariectomized under ketamine anesthesia with buprenorphine-SR as an analgesic. Mice were allowed to rest for 2 weeks to dissipate all endogenous ovarian hormones. After the resting period, mice were injected with 100 ng of estrogen (E2; Sigma-Aldrich) dissolved in 100 µL of sesame oil on two consecutive days and then allowed to rest for 2 days. At this point, mice were randomly

divided into three groups of five: vehicle-treated (E2 priming) mice received four consecutive days of sesame oil injections; E2 group mice received 3 days of sesame oil injections followed by a single injection of 50 ng of E2 on the fourth day; the E2/P4 mice received 1 mg of progesterone (P4; Sigma-Aldrich) for three consecutive days followed by a single injection of 1 mg P4 plus 50 ng E2 on the fourth day. All hormones were delivered by subcutaneous injection in a 90:10 ratio of sesame oil:ethanol. Mice were euthanized 16 hr after the final hormone injection to collect the uteri. A small piece of tissue from one uterine horn was processed in 4% neutral-buffered paraformaldehyde for histology, and the remaining tissue was snap-frozen and stored at –80°C.

## Hormone analysis

For serum hormone level measurement, blood was collected from D4 pregnant mice before mice were sacrificed. Serum was separated from the blood by centrifugation and stored at –80°C before hormone analysis. Serum P4 and E2 levels were measured using ELISA kits (Enzo Life Sciences) according to the manufacturer's instructions.

## H&E staining

Tissues were fixed in 4% paraformaldehyde, embedded in paraffin, and then sectioned (5 µm) with a microtome (Leica Biosystem, Wetzlar, Germany). Tissue sections were deparaffinized, rehydrated, and stained with H&E as described previously (*Chadchan et al., 2019*). All the histology was performed on three sections from each tissue of individual mice, and one representative section image is shown in the respective figures.

## Histological analysis

For histological analysis, the collected tissues (oviduct or uteri) were fixed in 4% paraformaldehyde, embedded in paraffin, and sectioned. Sections (5 µm) were immunostained (n = 5 per group) as described previously (*Chadchan et al., 2019*). Briefly, after deparaffinization, sections were rehydrated in an ethanol gradient and then boiled for 20 min in citrate buffer (Vector Laboratories Inc, Newark, CA) for antigen retrieval. Endogenous peroxidase activity was quenched with Bloxall (Vector Laboratories Inc), and tissues were blocked with 2.5% goat serum in Phosphate-buffered saline and Bovine serum albumin (PBS) for 1 hr (Vector Laboratories Inc). After washing in PBS three times, tissue sections were incubated overnight at 4°C in 2.5% goat serum containing the primary antibodies listed in *Supplementary file 2*. Sections were incubated for 1 hr with biotinylated secondary antibody, washed, and incubated for 45 min with ABC reagent (Vector Laboratories Inc). Color was developed with 3, 3'-diaminobenzidine (DAB) peroxidase substrate (Vector Laboratories Inc), and sections were counter-stained with hematoxylin. Finally, sections were dehydrated and mounted in Permount histological mounting medium (Thermo Fisher Scientific, Waltham, MA).

## Transmission electron microscopy

For ultrastructural analysis, oviducts were fixed in 2% paraformaldehyde/2.5% glutaraldehyde (Ted Pella Inc, Redding, CA) in 100 mM cacodylate buffer, pH 7.2 for 1 hr at room temperature and then overnight at 4°C. Samples were washed in cacodylate buffer and postfixed in 1% osmium tetroxide (Ted Pella Inc) for 1 hr. Samples were then rinsed extensively in dH$_2$O prior to en bloc staining with 1% aqueous uranyl acetate (Ted Pella Inc) for 1 hr. Following several rinses in dH$_2$O, samples were dehydrated in a graded series of ethanol and embedded in Eponate 12 resin (Ted Pella Inc). For initial evaluation, semithin sections (0.5 µm) were cut with a Leica Ultracut UCT7 ultramicrotome (Leica Microsystems Inc, Bannockburn, IL) and stained with toluidine blue. Sections of 95 nm were then cut and stained with uranyl acetate and lead citrate and viewed on a JEOL 1200 EX II transmission electron microscope (JEOL USA Inc, Peabody, MA). Images at magnifications of ×3000–30,000 were taken with an AMT 8-megapixel digital camera (Advanced Microscopy Techniques, Woburn, MA).

## Immunofluorescence analysis

Formalin-fixed and paraffin-embedded sections were deparaffinized in xylene, rehydrated in an ethanol gradient, and boiled in a citrate buffer (Vector Laboratories Inc) for antigen retrieval. After blocking with 2.5% goat serum in PBS (Vector Laboratories) for 1 hr at room temperature, sections were incubated overnight at 4°C with primary antibodies (*Supplementary file 2*) diluted in 2.5%

normal goat serum. After washing with PBS, sections were incubated with Alexa Fluor 488-conjugated secondary antibodies (Life Technologies, Carlsbad, CA) for 1 hr at room temperature, washed, and mounted with ProLong Gold Antifade Mountant with DAPI (Thermo Fisher Scientific). All immunofluorescence images were obtained using a Zeiss LSM 880 confocal microscope (×10 and ×40 objective lens).

Fresh oviduct tissues from 8-week-old mice from control (n = 3) and cKO (n = 3) were collected and fixed in 4% PFA at 4°C overnight. Then, oviducts were washed with PBS and cryoprotected using gradually increased sucrose concentrations in a row (15 and 30% w/v). 24 hr later, oviducts were embedded in an OCT medium, frozen (−80°C), and cryo-sectioned in the CM3050S Cryostat (Leica Biosystems, Germany). The cryosections were mounted on Superfrost plus glass slides (Fisher Scientific, Pittsburgh, PA) and stored at −80°C until used.

Cryo-sections were processed for immunofluorescence staining as described before (*Chojnacki et al., 2023*). Briefly, cryosections were blocked in 2.5% normal goat serum. After washing with PBS, sections were blocked with Mouse on Mouse (M.O.M.) IgG blocking reagent (M.O.M. Fluorescein Kit, Vector Laboratories, #FMK-2201) diluted in 1% BSA per the manufacturer's instructions. After blocking, sections were incubated overnight at 4°C with primary antibodies (*Supplementary file 2*) diluted in 2.5% normal goat serum. Next day, following washing with PBS, sections were incubated with Alexa Fluor 488-conjugated secondary antibodies (Life Technologies) for 1 hr at room temperature, washed, and mounted with ProLong Gold Antifade Mountant with DAPI (Thermo Fisher Scientific). All immunofluorescence images were imaged using a Nikon Fluorescent microscope.

## Western blotting

Protein lysates (40 µg per lane) from uteri or oviducts were loaded on a 4–15% SDS-PAGE gel (Bio-Rad, Hercules, CA), separated in 1X Tris-Glycine Buffer (Bio-Rad), and transferred to PVDF membranes via a wet electro-blotting system (Bio-Rad), all according to the manufacturer's directions (*Kommagani et al., 2016*). PVDF membranes were blocked for 1 hr in 5% non-fat milk in Tris-buffered saline containing 0.1% Tween-20 (TBS-T, Bio-Rad), then incubated overnight at 4°C with antibodies listed in *Supplementary file 2* in 5% BSA in TBS-T. Blots were then probed with anti-rabbit IgG conjugated with horseradish peroxidase (1:5000, Cell Signaling Technology, Danvers, MA) in 5% BSA in TBS-T for 1 hr at room temperature. Signal was detected with the Pierce ECL Western Blotting Substrate (Millipore, Billerica, MA), and blot images were collected with a Bio-Rad ChemiDoc imaging system.

## RNA isolation and quantitative real-time RT-PCR analysis

Tissues/cells were lysed in RNA lysis buffer, and total RNA was extracted with the Purelink RNA mini kit (Invitrogen, Carlsbad, CA) according to the manufacturer's instructions. RNA was quantified with a Nano-Drop 2000 (Thermo Fisher Scientific). Then, 1 µg of RNA was reverse transcribed with the High-Capacity cDNA Reverse Transcription Kit (Thermo Fisher Scientific). The amplified cDNA was diluted to 10 ng/µL, and qRT-PCR was performed with primers listed in *Supplementary file 1* and TaqMan 2X master mix (Applied Biosystems/Life Technologies, Grand Island, NY) on a 7500 Fast Real-time PCR system (Applied Biosystems/Life Technologies). The delta-delta cycle threshold method was used to normalize expression to the reference gene 18S.

## Treatment of mice with polyphyllin VI

Polyphyllin VI (Selleck Chemicals, Houston, TX), a pharmacological agent that induces caspase-1-mediated pyroptosis, was used to study its effects on embryo transport in the oviduct (*Teng et al., 2020*). Eight-week-old C57BL/6 mice were injected for three consecutive days starting from 1 dpc with polyphyllin VI activator dissolved in 40% PEG-300 (15 mg/kg body weight). Dimethyl sulfoxide with 40% PEG-300 was administered as a vehicle.

## Statistics

A two-tailed paired Student *t*-test was used to analyze data from experiments with two experimental groups and one-way ANOVA followed by Tukey's post hoc multiple range test was used for multiple comparisons. All data are presented as mean ± SEM. GraphPad Prism 9 software was used for all statistical analyses. Statistical tests, including p values, are reported in the corresponding figure legends or, when possible, directly on the data image. To ensure the reproducibility of our findings, experiments

were replicated in a minimum of three independent samples, to demonstrate biological significance, and at least three independent times to ensure technical and experimental rigor and reproducibility.

## Acknowledgements

We thank Goutham Davuluri and Ashirbad Guria from our group for the technical assistance. We thank Dr. Robert Lawrence (senior editor at Baylor College of Medicine) for assistance with manuscript editing. We thank WashU Molecular Microbiology Imaging Facility for assisting us with transmission electron microscopy. This work was funded, in part, by the National Institutes of Health/National Institute of Child Health and Human Development (grants R01HD102680, R01HD104813, and R01HD065435) to RK, R01 HD-042311 to JPL, and in part by National Institutes of Health/National Heart, Lung, and Blood Institute (grant R35HL145242) to MJH.

## Additional information

### Funding

| Funder | Grant reference number | Author |
|---|---|---|
| Eunice Kennedy Shriver National Institute of Child Health and Human Development | R01HD102680 | Ramakrishna Kommagani |
| Eunice Kennedy Shriver National Institute of Child Health and Human Development | R01HD104813 | Ramakrishna Kommagani |
| Eunice Kennedy Shriver National Institute of Child Health and Human Development | R01HD065435 | Ramakrishna Kommagani |
| Eunice Kennedy Shriver National Institute of Child Health and Human Development | R01 HD-042311 | John P Lydon |
| National Heart, Lung, and Blood Institute | R35HL145242 | Michael J Holtzman |

The funders had no role in study design, data collection and interpretation, or the decision to submit the work for publication.

### Author contributions

Pooja Popli, Validation, Investigation, Visualization, Methodology, Writing – original draft, Writing – review and editing; Arin K Oestreich, Investigation, Methodology; Vineet K Maurya, Marina N Rowen, Validation, Investigation, Methodology; Yong Zhang, Michael J Holtzman, John P Lydon, Shizuo Akira, Kelle Moley, Resources; Ramya Masand, Writing – review and editing; Ramakrishna Kommagani, Conceptualization, Formal analysis, Funding acquisition, Methodology, Writing – review and editing

### Author ORCIDs

Pooja Popli ⓘ https://orcid.org/0000-0002-5334-2755
Ramakrishna Kommagani ⓘ https://orcid.org/0000-0003-0403-0971

### Ethics

All animal studies were approved by the Institutional Animal Care and Use Committee of Washington University School of Medicine, Saint Louis, MO, USA, and the Use Committee of Baylor College of Medicine, Houston, TX, USA. Animals were handled according to an approved institutional animal care and use committee (IACUC) of Baylor college of medicine with protocol number [AN-8890] and of Washington University in St. Louis with protocol number (number 20160227). All transgenic mice were maintained on a C57BL/6 genetic background (The Jackson Laboratory, Bar Harbor, ME) to

minimize variation in the gestation length. All experimental animals were housed 5 per cage in institutional animal facility in standard ventilated cages with free access to water and food and under a 12-hr light and dark cycle.

Reviewer #1 (Public review): https://doi.org/10.7554/eLife.97325.4.sa1
Reviewer #2 (Public review): https://doi.org/10.7554/eLife.97325.4.sa2
Reviewer #3 (Public review): https://doi.org/10.7554/eLife.97325.4.sa3
Author response https://doi.org/10.7554/eLife.97325.4.sa4

# Additional files

## Supplementary files
MDAR checklist

Supplementary file 1. List of primers and TaqMan probes.

Supplementary file 2. List of antibodies.

## Data availability
No dataset has been generated or analyzed in this study. All other relevant data are available in the main text or the supplementary materials.

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
