## [Editor Report · eLife Assessment]

This **important** study reports a novel function of ATG14 in preventing pyroptosis and inflammation in oviduct cells, thus allowing smooth transport of the early embryo to the uterus and implantation. The data supporting the main conclusion are **convincing**. This work will be of interest to reproductive biologists and physicians practicing reproductive medicine.

---

## [Referee Report · Reviewer #1 (Public review)]

This study by Popli et al. evaluated the function of Atg14, an autophagy protein, in reproductive function using a conditional knockout mouse model. The authors showed that female mice lacking Atg14 were infertile partly due to defective embryo transport function of the oviduct and faulty uterine receptivity and decidualization using PgrCre/+;Atg14f/f mice. The findings from this work are exciting and novel. The authors demonstrated that a loss of Atg14 led to an excessive pyroptosis in the oviductal epithelial cells that compromises cellular integrity and structure, impeding the transport function of the oviduct. In addition, the authors use both genetic and pharmacological approaches to test the hypothesis. Therefore, the findings from this study are high-impact and likely reproducible.

Comments on revisions: Thank you for your time revising the manuscript. The authors have addressed all of my previous concerns.

---

## [Referee Report · Reviewer #2 (Public review)]

In this manuscript, Popli et al investigated the roles of autophagy related gene, Atg14, in the female reproductive tract (FRT) using conditional knockout mouse models. By ablation of Atg14 in both oviduct and uterus with PR-Cre (Atg14 cKO), authors discovered that such females are completely infertile. They went on to show that Atg14 cKO females have impaired embryo implantation as well as embryo transport from oviduct to uterus. Further analysis showed that Atg14 cKO leads to increased pyroptosis in oviduct, which disrupts oviduct epithelial integrity and leads to obstructive oviduct lumen and impaired embryo transport. Authors concluded that Atg14 is critical for maintaining the oviduct homeostasis and keeping the inflammation under check to enable proper embryo transport.

Comments on revisions: Authors have addressed all my concerns in this revised version, which is substantial improved compared to the original version. I have no further comments.

---

## [Referee Report · Reviewer #3 (Public review)]

The manuscript by Pooja Popli and co-authors tested importance of Atg14 in female reproductive tract by conditionally deleting Atg14 use PrCre and also Foxj1cre. The authors showed that loss of Atg14 leads to infertility due to retention of embryos within the oviduct. The authors further concluded that the retention of embryos within the oviduct is due to pyroptosis in oviduct cells leading to defective cellular integrity. This revised version of the manuscript has addressed the remaining concerns that were raised earlier. The manuscript is now a convincing one.

---

## [Author Response]

The following is the authors’ response to the previous reviews.

**Public Reviews:**

**Reviewer #1 (Public review):**
This study by Popli et al. evaluated the function of Atg14, an autophagy protein, in reproductive function using a conditional knockout mouse model. The authors showed that female mice lacking Atg14 were infertile partly due to defective embryo transport function of the oviduct and faulty uterine receptivity and decidualization using PgrCre/+;Atg14f/f mice. The findings from this work are exciting and novel. The authors demonstrated that a loss of Atg14 led to an excessive pyroptosis in the oviductal epithelial cells that compromises cellular integrity and structure, impeding the transport function of the oviduct. In addition, the authors use both genetic and pharmacological approaches to test the hypothesis. Therefore, the findings from this study are high-impact and likely reproducible. However, there are multiple major concerns that need to be addressed to improve the quality of the work.Thank you for the additional data that solidified the conclusion of this study. The authors addressed almost all of my previous concerns in this revised manuscript. However, some key points wording still need to be addressed.Comments on revisions:In Fig. 2A, please ensure that these are 5.0 dpc samples since implantation has already occurred at this point. However, the embryo appeared free-floating adjacent to the luminal epithelial cells (LE), even in control.

We understand the reviewer’s concern. We have now replaced the previous H & E image with a clearer, higher-quality section that shows a fully attached embryo within a closed uterine lumen representing a typical implantation morphology at the D5 stage of pregnancy. (Revised Figure 2A)

Fig. 3A-B: "Approximately 80-90% of blastocysts" contradicts the quantification in Figure 3C, which showed a percentage of blastocysts below 50%. Please clarify and correct as needed.

In Fig. 3A-B, we mean to say approximately 80-90% embryos. We have now corrected the statement in the revised manuscript (Line no: 349-351).

The authors showed that Acetylated a-tubulin was present in the ampulla region of cKO (Fig. 4A). However, the revised manuscript still stated that (lines 397-399) ...there was a substantial loss of the ciliary epithelial cells (indicated by fewer a-tubulin and FOXJ1-positive cells) (Fig. 4B, left panel and Fig. S3)... So, the authors may want to tone down their conclusion regarding a "substantial loss" of ciliated epithelial cells if the quantification of ciliated cell number is not performed.

We thank the reviewer for this suggestion. To avoid redundancy and ambiguity, we have revised the statement as below (Line no: 391-395):

“As shown in Fig. 4A, normal ciliary structures were observed in the ampulla of both control and cKO oviducts. However, in the isthmus of cKO oviducts, we observed a reduction in both the FOXJ1- and PAX8-expressing cells (Fig. 4B, and Fig. S3).”

Fig. 4C - the areas with red inset boxes labeled for isthmus are not really isthmus (in both control and cKO). The zoomed-in images Fig. 4C - The far-right panel for both control and cKO, images are the transitional zone from the ampulla to the isthmus. The isthmus areas should have a thick muscle layer with almost no ciliated cells - see Fig. 4B cKO - those are true isthmus areas.

We thank the reviewer for noting this. We have corrected the label accordingly. Since ciliary epithelial cells predominantly reside in the ampulla, we have included high-resolution images specifically for the ampulla regions.

• Fig. 3A and 3C, it appears that the images were taken at different magnifications, but the scale bars are the same at 200 um. The authors, please double-check the scale bars.

We thank the reviewer for noting this. We have double-checked all the figures to ensure the scale bars are correctly displayed and aligned with the resolution.

• Fig. 6D - why polyphillin-treated samples did not sum to 100%? - please double-check.

Since approximately 50% of the embryos were retained in the oviduct following polyphyllin treatment (Figure 6C, upper bar), the bar in Figure 6D represents this percentage (50% retained) rather than 100%.

**Reviewer #2 (Public review)**
In this manuscript, Popli et al investigated the roles of autophagy-related gene, Atg14, in the female reproductive tract (FRT) using conditional knockout mouse models. By ablation of Atg14 in both oviduct and uterus with PR-Cre (Atg14 cKO), authors discovered that such females are completely infertile. They went on to show that Atg14 cKO females have impaired embryo implantation as well as embryo transport from oviduct to uterus. Further analysis showed that Atg14 cKO leads to increased pyroptosis in oviduct, which disrupts oviduct epithelial integrity and leads to obstructive oviduct lumen and impaired embryo transport. The authors concluded that Atg14 is critical for maintaining the oviduct homeostasis and keeping the inflammation under check to enable proper embryo transport.The authors have barely addressed most of my concerns in this revised version with a few minor issues remaining to be addressed:(1) The authors tried to address my first concern regarding the statement that "autophagy is critical for maintaining the oviduct homeostasis". The revised statement in Lines 53-54 "we report that Atg14-dependent autophagy plays a crucial role in maintaining..." is still not correct. It should be corrected as " we report that autophagy-related protein Atg14 plays a crucial role in maintaining...".

We thank the reviewer for this nice suggestion. We have now modified the statement as suggested (Line no: 54).

(2) Line 349-351 described 80-90% of blastocysts retrieved from oviducts of cKO mice, which is in consistent with Figure 3B (showing more than 98%).

We thank the reviewer for noting this. We have now corrected the statement as: “Unexpectedly, oviduct flushing from cKO mice resulted in the retrieval of approximately 90% of embryos, suggesting their potential entrapment within the oviducts, impeding their transit to the uterus”. (Line No: 349-351).

(3) Line 447, "Fig. 5E" should be Fig. 6A. In addition, grammar error in the next sentence.

We have corrected the figure number and addressed the grammatical error.

(4) In Figure 6D, why the composition of blastocysts in chemical treated group do not add up to 100%.

As explained in Reviewer 1 responses, the bar in Figure 6D represents the 50% retained embryos from Figure 6C upper bar the full count.

**Reviewer #3 (Public review):**
Summary:The manuscript by Pooja Popli and co-authors tested the importance of Atg14 in the female reproductive tract by conditionally deleting Atg14 use PrCre and also Foxj1cre. The authors showed that loss of Atg14 leads to infertility due to the retention of embryos within the oviduct. The authors further concluded that the retention of embryos within the oviduct is due to pyroptosis in oviduct cells leading to defective cellular integrity. The revised manuscript has included new experimental data (Figs. S2B, 5B, 5C, and S3) that satisfied the concerns of this reviewer. The manuscript should provide important advancement to the field.

We sincerely thank the reviewer for the thoughtful evaluation of our manuscript and appreciate your constructive feedback.